# Characterization of Extracellular Vesicle Cargo in Sjögren’s Syndrome through a SWATH-MS Proteomics Approach

**DOI:** 10.3390/ijms22094864

**Published:** 2021-05-04

**Authors:** Francesco Finamore, Antonella Cecchettini, Elisa Ceccherini, Giovanni Signore, Francesco Ferro, Silvia Rocchiccioli, Chiara Baldini

**Affiliations:** 1Clinical Phisiology Institute-CNR, 56124 Pisa, Italy; francesco.finamore.1983@gmail.com (F.F.); ceccherini@ifc.cnr.it (E.C.); silvia.rocchiccioli@ifc.cnr.it (S.R.); 2Department of Clinical and Experimental Medicine, Rheumatology Unit, University of Pisa, 56126 Pisa, Italy; francescoferrodoc@gmail.com (F.F.); chiara.baldini@unipi.it (C.B.); 3Fondazione Pisana per la Scienza, S Giuliano Terme, 56017 Pisa, Italy; g.signore@fpscience.it

**Keywords:** Primary Sjögren’s syndrome, salivary proteomics, mass spectrometry, SWATH-MS

## Abstract

Primary Sjögren’s syndrome (pSS) is a complex heterogeneous disease characterized by a wide spectrum of glandular and extra-glandular manifestations. In this pilot study, a SWATH-MS approach was used to monitor extracellular vesicles-enriched saliva (EVs) sub-proteome in pSS patients, to compare it with whole saliva (WS) proteome, and assess differential expressed proteins between pSS and healthy control EVs samples. Comparison between EVs and WS led to the characterization of compartment-specific proteins with a moderate degree of overlap. A total of 290 proteins were identified and quantified in EVs from healthy and pSS patients. Among those, 121 proteins were found to be differentially expressed in pSS, 82% were found to be upregulated, and 18% downregulated in pSS samples. The most representative functional pathways associated to the protein networks were related to immune-innate response, including several members of S100 protein family, annexin A2, resistin, serpin peptidase inhibitors, azurocidin, and CD14 monocyte differentiation antigen. Our results highlight the usefulness of EVs for the discovery of novel salivary-omic biomarkers and open novel perspectives in pSS for the identification of proteins of clinical relevance that could be used not only for the disease diagnosis but also to improve patients’ stratification and treatment-monitoring. Data are available via ProteomeXchange with identifier PXD025649.

## 1. Introduction

Primary Sjögren’s syndrome (pSS) is a complex heterogeneous disease characterized by a wide spectrum of glandular and extra-glandular manifestations, potentially leading to parotid MALT lymphoma [1]. Although the involvement of salivary and lachrymal glands is the hallmark of the disease, during pSS progression any organ and system can be involved including lungs, kidneys, liver, nervous, and musculoskeletal system [2]. Thus, the spectrum of the disease can vary from a benign slowly progressive autoimmune exocrinopathy to a severe systemic disorder with remarkable clinical heterogeneity and scattered complications [3]. Notably, one-fifth of pSS patients may present major organ involvement with potentially severe end-organ damage and five percent of patients may also develop non-Hodgkin’s lymphoma [4]. Limited knowledge of the mechanisms of pSS disease variants has possibly represented the greatest obstacle in improving patients’ diagnosis, early stratification and personalized treatment, particularly explaining the failure of the ongoing “one size-fits all therapy” [5]. In other words, it is increasingly clear that pSS is a complex disease made up of number of disease subsets that deserves tailored approaches to avoid under- and/or over-treatments [6]. This is of particular importance, since the last years with the emergence of targeted therapies, used also for pSS, there is an urgent need to define novel and reliable biomarkers able to improve the definition of disease activity, progression, damage, and response to treatment and eventually, to implement precision medicine on the basis of individual’s disease expression [7]. During the last decade, salivary proteomics has appeared as a promising tool for identifying specific biomarkers able to mirror no-invasively pSS-related salivary gland inflammation and dysfunction [8]. Indeed, existing literature has described a number of qualitative and quantitative protein abnormalities in pSS saliva composition potentially correlated with the oral inflammation and salivary rheological alterations observed during the disease course suggesting that salivary proteomic biomarkers and salivary cytokines may be able to reflect not only the local but also the systemic activity of the disease [9,10,11,12,13,14,15]. However, despite the advantages recognized in studying whole saliva (WS), several potential limitations have been gradually come to light. Among the others, the presence of high abundant proteins in WS that might mask less represented proteins. Recently, a growing interest has arisen for extracellular vesicles-enriched saliva (EVs) as promising source of more preserved putative biomarkers. EVs have been extensively analyzed in pSS [16,17] to characterize the expression of small non-coding RNA molecules (miRNAs). However, little is known about the protein composition of salivary EVs in pSS, nor to what extent their content may disclose pathogenetic pathways involved in the disease development and progression. Therefore, in this pilot study, we used a sequential window acquisition of all the theoretical fragment ion spectra (SWATH-MS) approach to monitor the dynamics of EVs sub-proteome in pSS salivary samples in order to assess their value as a source of disease-related biomarkers in comparison with WS. The advantages in using this data-independent acquisition method rely on its unique feature to combine the deep proteome coverage capabilities of typical shotgun proteomics with accurate quantification of targeted proteomics without suffering of their own limitations (a random and irreproducible precursor ion selection that lead to under-sampling and the lack in quantifying large fractions of a proteome due to the limited number of transitions to measure, respectively). We specifically focused on EV enriched saliva and, besides the assessment of differential expressed proteins between pSS and healthy control EVs samples, a co-expression network analysis (WGCNA) was adopted to identify those groups of proteins with similar expression patterns that may be functionally related and associated with pSS. These key proteins could be of clinical importance as diagnostic biomarkers or therapeutic targets.

## 2. Results

### 2.1. Comparison between Whole Saliva and Salivary EVs Protein Profiles

We firstly analyzed and characterized vesicle distribution in EV enriched samples. Upon analysis with DLS (Appendix A), we identified three main vesicle distributions (69 ± 24 nm, 330 ± 150 nm, and 5307 ± 395 nm, with relative intensity 61.7%, 33.7%, and 2.9%, respectively). This heterogeneous distribution was expected since no attempt to specifically retrieve and isolate SEVs had been made, with the aim to preserve the full molecular information retrievable from biologic specimen. Note that the multimodal dispersion obtained with this approach did not allow a complete characterization of each single vesicle population in the sample. Qualitative/quantitative analysis of proteins extracted from WS and EV enriched saliva of five healthy controls was carried out by SWATH-MS to investigate on the different proteomics profiles of these two biological compartments. A total of 278 proteins were identified and quantified from EVs with an average coefficient of variation of 8.8% among each technical replicate; 171 proteins were quantified from WS with an average coefficient of variation lower than 12.06% between each technical replicate. The EVs dataset was compared with Vesiclepedia and ExoCarta (TOP100 most common human EV proteins) databases and 35 proteins resulted comprised in this database (Appendix A). Overall, EVs and WS shared 163 proteins whereas 115 and 8 were uniquely found in EVs and WS, respectively (Figure 1A).

Proteins commonly identified/quantified in both EVs and WS included groups of proteins previously found in saliva like cystatins family (B, D, S, SA, and SN), BPI fold-containing family A and B members, mucins 5AC and CB and mucin 7, some members of S100 proteins (A8 and A9) and prolactin-inducible protein that was previously shown as potential salivary biomarker of pSS by our group [9]. Proteins exclusively quantified in WS mainly included a class of proteins that are major components of parotid and submandibular gland tissue in humans which are the basic salivary proline-rich protein 2, small proline-rich proteins 2A, 2B, 2D, 2E, 2G, and 2F. On the other hand, proteins involved in extracellular exosome biology like several ADP ribosylation factors (ARF1, 3, and 5), some annexins (A2, A3, and A6), 7 members of RAB proteins (RAB1A, 1B, 8A, 8B, 10, 13, and 15) involved in vesicular trafficking and protein involved in immune response like azurocidin, S100 proteins (S100A11, A12 and E), MIF and bactericidal permeability-increasing protein (BPI) were exclusively found in EVs. In order to have a comprehensive view of the main differences in protein profiles between EVs and WS subsets, a GO enrichment analysis was performed for cellular components (Figure 1B). Results evidenced meaningful differences in GO terms between EVs and WS with a small percentage (27%) of shared features that, however, show significant differences. In particular, EVs samples were found to be largely enriched in extracellular exosome proteins, in proteins belonging to cytoplasmic vesicles and secretory vesicles, lysosomes, proteins involved in vesicle transport between endoplasmic reticulum and Golgi apparatus, to cell membrane and structural proteins of phagocytic and endocytic vesicles membranes. Proteins associated to immunoglobulin complex, hemoglobin complex, MHC protein complex, and components of extracellular matrix like collagen were instead to be uniquely found in WS samples. Moreover, the abundance of those proteins that were found to be in common between EVs and WS were compared in order to figure out their likelihood expression similarity. Results showed an extremely low correlation between EVs and WS (squared Pearson correlation coefficient R^2^ = 0.27) (data not shown).

### 2.2. Identification of a Meta-Module of Co-Expressed Proteins Related to an Inflammatory Phenotype of Sjögren’s Syndrome

The overall dataset of normalized protein abundances from pSS and healthy controls (Appendix A) obtained through SWATH-MS was used to study the relationship between the correlation patterns of protein groups that show similar expression levels (modules) and pSS. Using WGCNA analysis we identify four main protein modules (showed in different colors) generated in the hierarchical clustering from 12 samples across patients and controls (Figure 2A).

The module with the higher number of proteins was the turquoise module (72 proteins) followed by the blue and brown modules with 58 and 53 proteins, respectively and finally the yellow module with 30 proteins. Grey module contains those proteins that were not associated to any other module in the analysis. In order to evaluate the degree of interconnectivity between each module and the association between modules and pSS as the clinical trait of interest, the module network dendrogram was generated by clustering module distances (a measure of module dissimilarity). A correlation higher than 75% (dissimilarity, 0.25) was used to define those groups of modules that show a high interconnectedness between them (meta-modules). As shown in the dendrogram, blue and brown modules are highly related, but their mutual correlation is stronger than their correlation with pSS. On the other hand, yellow module and turquoise module are correlated but the latter showed a more significant correlation with pSS trait, as they are tightly clustered together as a different meta-module separated from the blue and brown meta-module (Figure 2B). To further quantify co-expression similarity among modules, the adjacency on inter-module correlation was calculated. The heatmap shows the module-module relationship in which the progressively more saturated blue and red colors indicate a high co-expression interconnectedness (Figure 2C). In our study, pSS trait showed a high adjacency with turquoise module but not with yellow, blue, and brown modules, indicating that despite turquoise and yellow modules were clustered together in the dendrogram, their protein expression levels are negatively correlated. In addition, brown and blue modules were neither clustered nor positively correlated with pSS trait, evidencing differences in the expression profiles of those set of modules. Moreover, among these modules, turquoise module showed the highest module significance (average protein significance across the module proteins) compared to the others, meaning that this module is more significantly related to pSS trait (Figure 2D). Since highly correlated proteins in a module may play important roles in biological processes, the whole 72 proteins of the turquoise module were extracted and used to generate a protein–protein interaction network of co-expressed proteins (Figure 2E). Most of the proteins within the network showed a fold change that ranges between 1.3 and 3.9 (color-coded in scale of yellow-violet) with an average value of 2.3 and a *p*-value between 0.04 and 5.3 × 10^−9^. The most representative functional pathways associated to the network proteins were related to immune innate response processes like neutrophil degranulation (*p* = 2.3 × 10^−22^), interleukin 12 signaling (*p* = 1.2 × 10^−18^), regulation of cytokine secretion (9.5 × 10^−11^) and antimicrobial humoral response (*p* = 2.0 × 10^−9^), to cite few (Figure 2F). These results are in line with the work hypothesis that inflammatory phenotype observed in pSS patients is also extended to salivary EVs ^10,17^.

### 2.3. Differential Expression Protein Analysis Reveals the Relationship between Sjogren’s Syndrome and Innate Immune Response

The impact of pSS in modulating protein expression profiles was assessed by comparing the EV protein cargoes between pSS patients and healthy controls. Significant and differentially expressed proteins were selected by performing a Mann–Whitney test, using a *p*-value < 0.05 and a fold change −1.5 > FC > +1.5. A total of 121 proteins were found to be differentially expressed corresponding to 42% of the total number of identified and quantified proteins (Appendix A). Among these 121 proteins, 82% and 18% were found to be up- and downregulated in EVs derived pSS samples (Figure 3A).

A functional protein-protein interaction network was constructed using up- and downregulated proteins and the most significant sub-network with the highest number of nodes (46) and edges (112) was extracted for further analysis (Figure 3B). The differentially expressed proteins within the top scoring sub-network were mapped onto biological pathways mainly related to innate immunity and inflammation with neutrophil degranulation (red color-coded) as the most representative and significant GO term (21 proteins, *p*-value = 2.5 × 10^−22^) (Figure 4A). The most predominant proteins involved in neutrophil degranulation were several members of S100 protein family (S100A7, A8, A9, A11, and A12), resistin (RETN), serpin peptidase inhibitors (SERPINB1, and SEPINB5), azurocidin (AZU1), and monocyte differentiation antigen (CD14), to cite few (Figure 4B).

Moreover, a significant enrichment in proteins involved in antimicrobial humoral response (8 proteins, *p*-value = 2.7 × 10^−8^), regulation of chemokine production (6 proteins, *p*-value = 4.1 × 10^−7^), leukocyte migration (8 proteins, *p*-value = 1.6 × 1^−7^) and IL−12-mediated signaling pathway (6 proteins, 1.2 × 10^−7^) was also observed (yellow color-coded). The latter is of importance since it has been previously shown the role of IL-12 and its downstream mediators in the onset and progression of pSS complications [18,19]. In this study, 10 specific proteins associated to IL-12 signaling pathway were identified and quantified, 8 of them were found to have similar expression levels and 6 out of 8 proteins were found to be differentially expressed. Overall, 6 proteins out of 10 were commonly detected by the co-expression network analysis and differential expression analysis, namely annexin A2 (ANXA2), cofilin-1 (CFL-1), plastin-2 (LCP1), macrophage migration inhibitory factor (MIF), protein S100 A8 (S100A8), and protein S100 A9 (S100A9) (Figure 4B). The remaining 2 proteins of the IL-12-mediated signaling pathway, moesin (MSN), and protein disulfide isomerase (P4HB), were not significant or below the selected fold change threshold. Other functional pathways with a significance level slightly below the ones mentioned above were related to macrophage activation (five proteins, *p*-value = 5.3 × 10^−6^), prostaglandin biosynthetic process (four proteins, *p*-value = 4.8 × 10^−6^), granulocyte chemotaxis (six proteins, *p*-value = 3.4 × 10^−6^), positive regulation of TNF production (five proteins, *p*-value = 5.1 × 10^−6^), and arachidonic acid secretion (four proteins, *p*-value = 6.6 × 10^−6^), to cite the most representative GO terms. Based on the co-expressed and differential proteins provided by our dataset, the hypothesis that salivary EVs may represent a meaningful inflammatory component in pSS phenotypic expression might be correct.

### 2.4. Immunometric Assessment of Specific Inflammatory Proteins in Sjogren’s Syndrome

The presence of specific proteins belonging to the inflammatory pathways evidenced with SWATH-MS and described above were investigated by Western blot. Among the increased-in-abundance proteins in pSS patients, we focused our attention on three representative molecules related to immune innate response (Annexin A2, S100-A8, S100-A9), as reported in Figure 4D.

## 3. Discussion

In this pilot study we aimed at investigating the sub-proteome of EVs in pSS searching for novel putative disease biomarkers. We first identified and quantified more than 100 proteins that were unique to EVs and not detected in WS. These proteins were largely enriched in extracellular exosome proteins, in proteins belonging to cytoplasmic vesicles and secretory vesicles, lysosomes, proteins involved in vesicle transport between endoplasmic reticulum, Golgi apparatus, and structural proteins of phagocytic and endocytic vesicles membranes. These data are in line with the amount of evidence on the active involvement of EVs in inter- and intra-cellular communications and that are looking at EVs as potential reservoirs of highly promising disease-specific biomarkers in pathological processes [20].

More specifically, in this study we identified a number of EVs candidate biomarkers apparently able to distinguish pSS patients from healthy volunteers. Here, we used the integration of two complementary approaches, namely co-expression and differential expression analysis, to screen potential protein biomarkers of SS that are biologically related. WGCNA identified a co-expression module related to SS. A total of 72 hub proteins within this module were characterized and 58 proteins were in common with the dataset of differential expressed proteins (Appendix A), meaning that these proteins may have a functional relation and can be used as likely biomarkers. These proteins, extracted and used to generate a protein-protein interaction network, appeared to play a major role in functional pathways related to immune innate response processes such as neutrophil degranulation, interleukin 12 signaling, regulation of cytokine secretion and antimicrobial humoral response. The most predominant differentially expressed proteins resulted several members of S100 protein family (S100A7, A8, A9, A11, and A12), resistin (RETN), serpin peptidase inhibitors (SERPINB1 and SEPINB5), azurocidin (AZU1), monocyte differentiation antigen (CD14), and many proteins belonging to the IL-12-mediated signaling pathway including annexin A2 (ANXA2), cofilin-1 (CFL-1), plastin-2 (LCP1), macrophage migration inhibitory factor (MIF), besides S100A8 and S100A9.

To date, despite the increasing number of studies aimed at exploring EV cargos in autoimmunity, limited data are available for pSS. Aqrawi et al. [10] have recently investigated salivary EVs in ten pSS patients. The authors found a number of significantly upregulated proteins in pSS; however, the DAVID analysis performed on these highly expressed proteins did not revealed any significantly affected signaling pathways involving cellular processes.

By contrast, our findings allowed us to focus on proteins that apparently play essential roles in functional pathways related to immune innate response and might be involved not only in the development of oral cavity inflammation but also in pSS systemic activity and progression.

We speculated that both the increased expression of proteins related to neutrophil activation and those belonging to IL-12-mediated signaling pathway might be particularly related to the oral mucosa health status and to the potential alterations of tongue and oral cavity related to pSS-dryness. Not surprisingly, these pathways have been described as abnormal also in chronic periodontitis as well as in oral squamous cell carcinoma. For example, azurocidin seems to have a protective role in alveolar bone loss during the early stages of periodontitis [21] whereas SERPINB1 positively correlated with cell migration and oral cancer metastasis [22]. Moreover, soluble monocyte differentiation antigen (CD14) has been described as a crucial factor influencing the susceptibility of human periodontal ligament stem cells to different pathogens and thus may contribute to the progression of periodontitis [23]. Plastin-2 (LCP1) regulates leukocyte adhesion to integrin and has been described among salivary proteins associated with periodontitis in patients with type 2 diabetes mellitus [24]. Finally, macrophage migration inhibitory factor (MIF), another important effector cytokine of the innate immune system, has been also associated with the disease severity in chronic periodontitis [25]. Noteworthy, none of our patients presented a clinically evident periodontitis thus we hypothesized that salivary EVs cargo may represent a useful source of early biomarkers able to reflect initial alterations of the gingival crevicular fluid and subclinical abnormalities of the oral cavity. These proteins may therefore be useful to assess the relationship between the innate immune system and oral manifestations in pSS, offering new therapeutic biomarkers to improve disease treatment options.

Moreover, and more intriguingly, we also observed the over-expression of EVs proteins that could be specifically related to pSS activity and progression. First of all, we confirmed the increased expression of S100A proteins (S100A7, A8, A9, A11, and A12). These proteins are essential ligands for the receptor for advanced glycation endproducts (RAGE) and act as damage-associated molecular pattern (DAMP) ligands to Toll-Like Receptors 4 (TLRs4) which in turn promote cellular and immunological abnormalities that foster systemic and local inflammation [26]. TLRs4 have shown significantly higher expressional levels in pSS salivary gland epithelial cells as well as in salivary-infiltrating mononuclear cells when compared to controls. Moreover, in cultured human salivary cells agonists to TLRs4 stimulate CD54 expression and IL-6 production through phosphorylation of MAPKs indicating that TLRs4 may contribute to pSS pathogenesis [27,28]. Noteworthy, S100A proteins have been described as over-expressed also in pSS sera and WS as closely associated with glandular and extra-glandular disease activity [9,26,29,30].

Similarly, we observed increased levels of resistin in pSS EVs subproteome. Resistin is an adipokine that influences the activity of various cells engaged in innate immune response and inflammatory processes mainly by affecting adhesion molecule expression, chemotaxis, apoptosis, and phagocytosis, as well as pro-inflammatory cytokines production and release [31]. Bostrom E.A. et al. demonstrated that resistin was upregulated in the minor salivary glands of patients with pSS; and that the levels of resistin were correlated to the intensity of glandular lymphocytic inflammation suggesting that resistin might represent a driving factor of focal sialoadenitis in pSS [32].

Besides these findings, we also found that both annexin-2 and cofilin-1 were increased in pSS EVs. Intriguingly, Cui Li et al. [33] found that both annexin 2 and cofilin-1 were over-expressed in parotid tissues of pSS patients, particularly in those patients with MALT/pSS. The authors also demonstrated that pSS patients with MALT lymphoma presented the highest levels of salivary autoantibodies anti-cophilin-1. Our results than seem to indicate that EVs may also provide potential candidate biomarkers of disease progression and lymphoproliferation in pSS.

This study has some potential limitations particularly related to the small number of patients enrolled. However, our results once validated, may open novel perspective on the usefulness of EVs for the discovery of novel salivary-omic biomarkers in pSS to be used not only for the disease diagnosis but also to improve patients’ stratification and treatment-monitoring.

## 4. Material and Methods

### 4.1. Patients

Seven patients (ages 25–75 yrs) with the diagnosis of pSS fulfilling the ACR/EULAR 2016 criteria [34] were recruited from the outpatient cohort at the Rheumatology Clinic of the University of Pisa. All patients complained for dry eye and dry mouth and presented reduced unstimulated salivary flow (mean 0.299 ± 0.225 mL/min) and pathological ocular tests results. All the patients presented also a positivity for the anti-Ro/SSA antibodies. Finally, histological evaluations of the minor salivary glands was performed in all the cases and the focus score ranged from 1.3 to 3.2. The control group comprised five healthy women ages 26–65 years with preserved unstimulated salivary flow rate (mean 0.467 ± 0.299 mL/min). None of the controls had any history of rheumatic disease, or subjective symptoms of ocular or oral dryness. At the time of saliva sampling, no patients or controls had a diagnosis of oral periodontitis. The study was approved by the Medical Research Ethics Committee of the University of Pisa. All the patients provided an informed consensus to participate to the study.

### 4.2. Saliva Collection

Unstimulated WS samples were collected early in the morning (between 8:00 and 10:00 a.m.) under standard conditions and all the subjects were asked to be on an empty stomach, without having had any drink or eaten any kind of food (including gum or candies) since the night before [35]. They rinsed mouth with water than spit into a 50 mL Falcon tube for 5 min. In order to minimize degradation of the proteins, the samples were processed immediately and kept on ice during the process. Between 1 and 7.5 mL of saliva was obtained from each pSS subject whereas volumes obtained from controls ranged from 2.5 to 10 mL.

### 4.3. Salivary EVs Purification and Sample Preparation for Mass Spectrometry-Based Proteomics

EVs were isolated from whole saliva (WS) of seven pSS patients and five healthy controls by differential centrifugation as follows: each sample was centrifuged at 300× *g* to eliminate cells, bacteria, and potential food debris.

Only for healthy controls, each saliva sample was split in two before any further processes to perform proteomics analysis on whole saliva and comparison of identified proteins with EV-enriched samples. Albumin and IgG were removed from WS specimens by immunoaffinity chromatography using the proteo-prep immunoaffinity albumin and IgG depletion kit (Sigma Aldrich, St. Louis, MA, USA).

For all the samples allocated for EV-enrichment, the supernatant was diluted to 10 mL with PBS and centrifuged at 2000× *g* for 30 min at 4 °C to remove apoptotic blebs, and eventually supernatant was diluted to 20 mL with PBS and subjected to ultracentrifugation at 100,000× *g* for 2 h at 4 °C to obtain pellets EVs [36]. WS and EV proteins were extracted with 50 mM Tris-HCl, 120 mM NaCl, 1% sodium deoxycholate. Briefly, samples were lysed through five repeated freeze-thaw cycles (freezing in liquid nitrogen for 30 s and thawing at 50 °C for 2 min), sonicated for 5 min (five cycles of 20 s with an interval between cycles of 40 s on ice) and then clarified by centrifugation at 16,000× *g* for 10 min at 4 °C. Protein concentration was determined by the bicinchoninic acid assay (Thermo Scientific, Rockford, IL, USA) using serum albumin as standard. For each condition, 100 µg of proteins were reduced with dithiothreitol (10 mM, for 30 min, at 65 °C) and alkylated using iodoacetamide (22 mM, 30 min, room temperature) in dark conditions. Protein digestion was performed using trypsin (*w*/*w* ratio 1:50) at 37 °C for 16 h. Samples were incubated with 1% trifluoroacetic acid (Sigma-Aldrich) for 45 min at 37 °C to quench trypsin reaction and to remove sodium deoxycholate by acid precipitation. Samples were centrifuged at 16,000× *g* for 10 min and subsequently desalted with Mobicol spin columns equipped with 10 µm pore size filters and filled with VersaFlash C18 spherical 70 Å silica particles (Sigma-Aldrich). Peptides were lyophilized and consequently dissolved in 2% acetonitrile/0.1 formic acid to achieve a final peptide concentration of 1 µg/µL before liquid chromatography–tandem MS (LC-MS/MS) analysis. Three replicate injections were carried out for each sample to assess technical reproducibility.

### 4.4. Dynamic Light Scattering Analysis

Measurements by DLS were performed at 25 °C in a 50 μL disposable cuvette on a Zetasizer nano-Z DLS (Malvern Instruments, Worcestershire, UK) following the manufacturer’s instructions. PBS solutions of vesicles were analyzed with a single-scattering angle of 173°. Each value reported is the average of three consecutive measurements (Appendix A).

### 4.5. MS Acquisitions: IDA and SWATH-MS

The equivalent of 5 µg per sample were directly loaded with an Eksigent expert™ microLC 200 system (Eksigent, AB Sciex, Framingham, MA, USA) and acquired using a 5600+ TripleTOF mass spectrometer (AB Sciex). After loading, peptides were separated on a Jupiter 150 × 0.3 mm, 4 µm 90 Å capillary column with a gradient from 2 to 35% buffer B (acetonitrile/0.1 formic acid) in 40 min at a flow rate of 5 µL/min. LC column was directly interfaced with a DuoSpray™ ESI ion source operating at 5.5 kV for peptide ionization. We used an information dependent acquisition (IDA) tandem mass spectrometry (MS) for spectral library generation that relies on a survey MS1 scan followed by the selection of a maximum of 20 most abundant precursor ions and their further fragmentation by collisional induced dissociation (CID) using nitrogen N_2_ as inert gas to generate MS2 spectra. MS1 survey and MS2 scans were acquired with a resolving power of 30,000 and 25,000 and over a mass range of 250–1250 *m/z* and 150–1500 *m/z*, respectively. Isolation width for precursor ion selection was set at 0.7 *m/z* on a Q1. The accumulation time was set to 250 milliseconds for MS1 scans while 100 milliseconds for MS2 scans. Charge states of 1+ were excluded from ion selection. Rolling collision energy with a collision energy spread across 5 eV and background subtraction were enabled to achieve the optimal fragmentation according to *m/z* ration and charge state and to increase sensitivity. WS samples and salivary EVs peptide samples from healthy controls and pSS patients were analyzed by a data-independent method based on sequential window acquisition of all the theoretical fragment ions spectra (SWATH-MS). SWATH acquisitions were performed over 40 overlapping isolation mass windows of variable length (min length 10 Da/mass selection with 1 Da of window overlap) depending on the peptide density distribution along the entire mass range of 400–1250 *m/z* (Appendix A). Precursor ion activation was performed by CID as described before. An accumulation time of 200 milliseconds for MS1 and 90 milliseconds for MS2 scans, resulted in an overall duty cycle of 3 s (≈ 8 points per elution peak). Maximum resolving power was achieved at 20,000 (FWHM) at 400 *m/z*.

### 4.6. Spectral Library Generation and Statistical Analysis

Spectral library was created by combining the outputs from data dependent acquisition (DDA) MS runs previously acquired using the IDA-MS method described above by pooling together pSS and healthy control EVs and WS samples. Four replicate injections were performed and the acquired raw data (Wiff, profile spectra) were converted into a peak list format (mzML, centroid spectra) using the ProteoWizard tool MSConvert [37]. Protein identification was carried out using both X!Tandem and Comet search tools through the TPP [38] (PMID:25631240) software suite to increase the robustness of the identification. Briefly, peak lists were searched against a reviewed human database (UniProtKB/Swiss-Prot, 20386 sequences, release May 2018) using a precursor ion and fragment ion tolerance of 20 and 50 ppm, respectively, a precursor charge state between 2+ and 5+ and a maximum number of 2 trypsin miss cleavages. Carbamidomethylation (+57.021 Da) of cysteine residues and oxidation (+15.995 Da) of methionine residues were chosen as fixed and variable modifications, respectively. Peptide spectrum matches (PSMs) (pepXML files from both X!Tandem and Comet search tools) were scored, combined and re-scored using PeptideProphet and iProphet to increase the confidence between correct and incorrect hits. False positives were filtered out with an FDR lower than 5% using MAYU software (corresponding to the lowest iProphet probability) and the resulting list of PSMs was converted into a redundant spectral library using SpectraST [39]. At this stage, the retention times of all the peptides were transformed into normalized retention times using 11 endogenous retention time reference peptides (iRTs). These iRT peptides cover the entire retention time range of the analyzed samples with intensity values above 10,000, a minimum signal-to-noise ratio of 5 and a coefficient of determination of r = 0.9977 (Appendix A). In order to increase the accuracy and consistency of the fragmentation pattern, MS2 spectrum entries corresponding to a redundant peptide sequence assignment were collapsed into a single consensus spectrum. Consensus spectral library was then converted into a SWATH assay library in which the following parameters were included: a minimum number of eight fragment ions per peptide precursor, fragment ions smaller than 350 and bigger than 2000 *m/z* were filtered out, only b- and y- ion types with charge states of 1+ and 2+ were considered and only fragments with a mass accuracy equal or below ±0.05 Th of the expected mass were used. Assay library (in table format and containing a total of 350 proteins) was uploaded into PeakView (AB Sciex, v2.2) alongside SWATH-MS raw data of 36 samples (7X3 technical replicates vs 5X3 technical replicates in each group) for subsequent peak group definition. Each peak group correspond to the signals of specific fragment ions derived from a target peptide integrated over their chromatographic elution time. Fragment ion signals/peptide precursor groups (transitions) areas were extracted for each assay library matched peptide and integrated together to obtain the peptide peak areas. Peak areas of peptides associated to unique proteins were summed together to achieve total protein abundance values that were normalized through the total ion current (TIC) extracted from the full MS1 survey scan acquisition for each run. Proteins were considered to be significantly different with a *p*-value lower than 0.05 and a fold change (|FC|) > 1.5.

### 4.7. Protein Co-Expression Network Analysis and Protein–Protein Interaction Network Construction

Co-expression network was generated using WGCNA (v1.69) package in R [40]. Normalized protein abundances from salivary EVs of healthy controls and pSS patients were used to calculate Pearson correlations between all pairs of proteins across all control and pSS samples. Correlation values were used to construct a matrix of adjacencies that was computed into a topological overlap matrix (TOM) where proteins with a high topological overlap (highly similar co-expression relationship) were grouped together into modules by hierarchical clustering analysis. Modules of highly interconnected groups of proteins were identified as branches on a dendrogram using the R Package Dynamic Tree Cut algorithm [41]. A soft-thresholding power of 12, for which the scale-free topology fit index reached 0.90, and a minimum module size of 30 proteins per module were chosen. Primary SS condition was used as our study trait to investigate on the correlation among protein expression modules and pSS phenotype. A minimum height of 0.25 was chosen to define those modules whose expression profiles was similar, and the modules were randomly color-labeled. The module with the highest significant correlation with the pSS trait (gene significance) was extracted and analyzed by gene ontology (GO) and pathway analysis.

### 4.8. Functional Network Generation and Gene Ontology (GO) Terms Enrichment

Functional protein-protein interaction networks were constructed using the STRING database and the most representative protein complexes were extracted according to their level of interconnectivity within a subnetwork. Briefly, highly connected regions (clusters) were extracted from the whole network using the Cytoscape plug-in MCODE [41] by applying a degree cut-off of 4 with no loops, a node score cut-off of 0.2, a k-score of 2 and a maximum depth of 100. Haircut function was enabled. GO analysis was carried out with the Cytoscape plug-in ClueGO [42]. GO parameters used for functional grouping (biological process, cellular component, and pathways) were as follows: a *p*-value ≤ 0.01 integrated with a Bonferroni step down correction, a GO tree interval between 4 and 8, a minimum number of genes per cluster of 4 with a 4% of genes, a kappa score of 0.4 and an initial group size of three terms with a percentage of overlapping terms per group of 50%. GO term fusion of similar associated genes was enabled.

### 4.9. Data Availability

Salivary EVs mass spectrometry proteomics raw data from SWATH-MS acquisitions of WS and pSS patients and healthy controls EVs, altogether with consensus spectral library and SWATH assay library have been deposited to the ProteomeXchange via the PRIDE [43] partner repository with the dataset identifier PXD025649.

### 4.10. Western Blot Analysis

Protein samples were loaded on polyacrylamide gel at the concentration of 10 μg/well and separated by SDS/PAGE, using 4–20% precast gel (Mini-PROTEAN^®^ Precast Gels, Bio-Rad, Hercules, CA, USA). Proteins were transferred on nitrocellulose membranes and immunoblotted by monoclonal antibodies anti-s100A8, anti-s100A9 and ANXA2 (R&D systems). Protein-bands were detected by chemiluminescent Pierce™ ECL Western blotting substrate (Bio-Rad) and analyzed by ChemiDoc™ XRS (Bio-Rad). Densitometric analysis was conducted using ImageJ software.

## Figures and Tables

**Figure 1 ijms-22-04864-f001:**
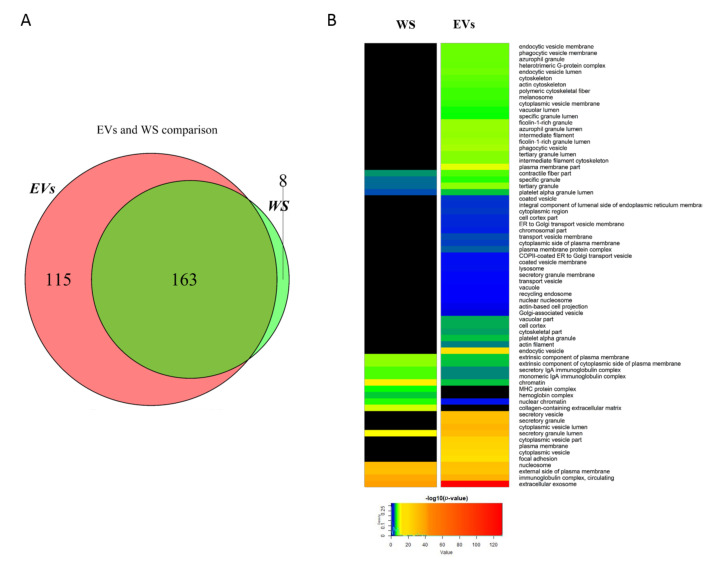
Qualitative comparison between whole saliva (WS) and extracellular vesicles (EVs) proteomics profiles. Venn diagram shows the differences in the number of identified proteins among the two groups of samples (**A**). GO cellular component (CC) analysis evidenced a significant enrichment of extracellular vesicles processes in EVs fraction compared to WS. Color-coded scale is associated to the significance of association of proteins to a specific GO term. The higher is the significance of association, the more the colors are shifted to red inks (**B**).

**Figure 2 ijms-22-04864-f002:**
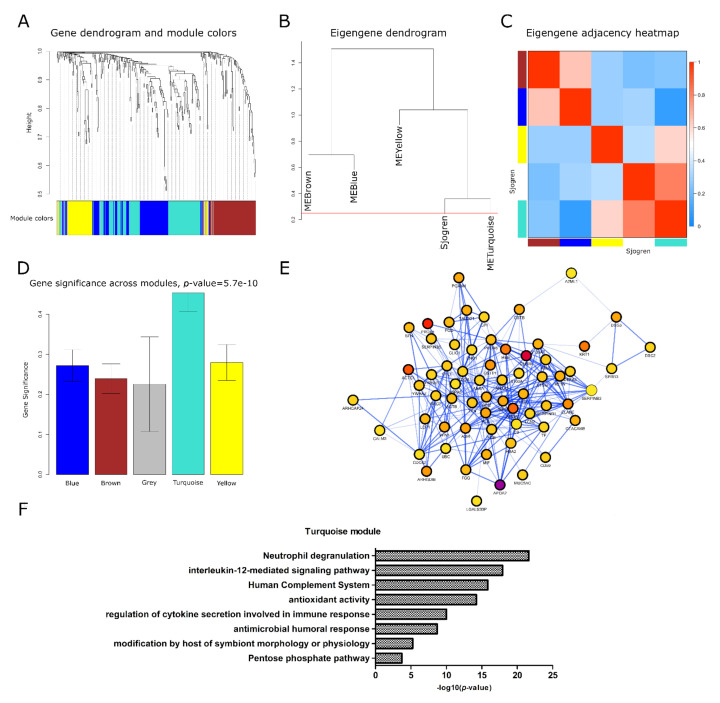
Protein co-expression network analysis of EVs Sjogren’s data in healthy controls and patients. Module dendrogram obtained by average linkage hierarchical clustering. The color row underneath the dendrogram shows the module assignment determined by the dynamic tree cut (**A**). Hierarchical clustering dendrogram of module proteins (labeled by their colors) and the sample trait (Sjogren’s). Branches of the dendrogram (the meta-modules) group together proteins that are positively correlated (**B**). Heatmap plot of the adjacencies in the module network including the Sjogren’s trait. Each row and column in the heatmap corresponds to each protein module (labeled by color) or Sjogren’s trait. In the heatmap, blue color represents low adjacency (negative correlation), while red represents high adjacency (positive correlation). 45 degree diagonal of red color correspond to the meta-modules (**C**). Barplot of mean protein significance across modules. The higher is the mean protein significance in a module, the more significantly related the module is to the clinical trait of interest (Sjogren’s trait) (**D**). Protein–protein interaction network of proteins associated to the turquoise module. Each node correspond to a specific protein and color nodes is related to the protein expression level (color-code range, min |FC|=1.1268 yellow, max |FC|=6.9088 purple) (**E**). Biological process terms associated to the network of co-expressed proteins. Bar length is related to the significance of proteins to each GO term (**F**).

**Figure 3 ijms-22-04864-f003:**
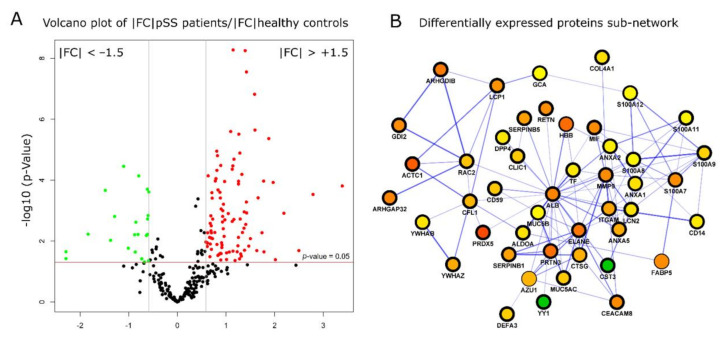
Differential expression analysis of EVs Sjogren’s data in healthy controls and patients. Volcano plot shows the significantly up- (red) and down- (green) regulated proteins with a fold change of FC = 1.5 and *p* < 0.05 (**A**). The most representative protein subnetwork was extracted from a comprehensive protein interaction network of differentially expressed proteins according to the number of nodes (proteins) and edges (links) within the subnetwork. Node colors indicate the fold change expressed as protein abundance ratio between Sjogren’s patients and controls groups (min |FC|=0.6714, green, max |FC| =10.4881, purple (**B**).

**Figure 4 ijms-22-04864-f004:**
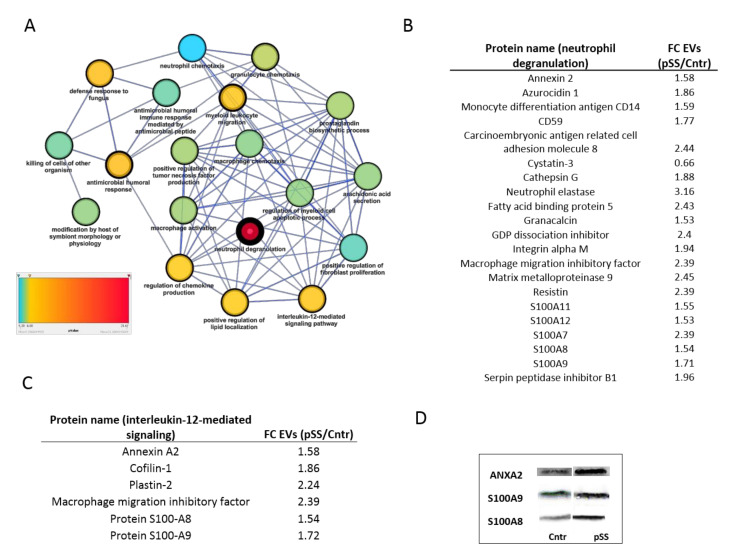
Most significant (*p* < 0.001) biological processes of protein subnetwork are shown as interconnected dots which color and width indicates the GO term significance and the number of proteins associated to each term, respectively (**A**). Differentially expressed protein found to be significantly up- or downregulated (*p* < 0.05, |FC| = 1.5) of the neutrophil degranulation (**B**) and IL-12-mediated signaling pathways (**C**) are shown. (**D**) Representative images of western blot analyses for AnnexinA2, S100A8 and S100A9.

## Data Availability

The data presented in this study are openly available in PRIDE repository with the dataset identifier PXD025649.

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
