# Peer review of "Characterization of Extracellular Vesicle Cargo in Sjögren’s Syndrome through a SWATH-MS Proteomics Approach"

_ijms, 2021, doi:10.3390/ijms22094864_

Round 1

Reviewer 1 Report

The overarching goal of this study is to characterise the proteome of salivary extracellular vesicles (EVs) and whole saliva derived primary sjogren’s syndrome (pSS) patients. The proteomes were assessed by SWATH-mass spectrometry, a data-independent mass spectrometry approach that allows for reliable and targeted proteomic analyses. SWATH-MS is growing as a useful proteomic method biomarker discovery as SWATH-MS theoretically sequences all analyses within a sample, and the data can be mined retrospectively as the spectral library grows. The study identified a number of differentially expressed proteins between pSS and healthy control EVs that were related to the immune response. This is an interesting study and holds immense value for developing EV-based biomarker panels that can be used to diagnose and monitor autoimmune disorders such as pSS.

The methodological approach for the manuscript is logical, however there are a few key points that should be addressed to improve the content of the manuscript. Furthermore, several passages would benefit from further refinement and clarification to improve the readability of the manuscript. Please refer to the major and minor points below.

Major points:

  • As this is a study centred around extracellular vesicles (EV), it is imperative that the authors follow the MISEV2018 guidelines for the characterisation of their salivary EV populations. Is there a reason that the authors have omitted from displaying an EV characterisation figure in this study? EV studies typically show the quality, size and content of their EV populations in a figure. Typically, such a figure would comprise of a size distribution profile determined by nanoparticle tracking, EV quality/integrity/size determined by transmission electron microscopy and the identification of EV-marker proteins determined by western blotting or mass spectrometric methods. I would recommend the authors to revise the MISEV2018 guidelines and provide a figure containing the appropriate characterisations for their isolated salivary EV populations.
    1. Based on these characterisations, it is recommended that the broad term of EVs is used and that EVs are categorised by their size ranges (eg., small-EVs, large EVs, etc) rather than exosomes/microparticles/apoptotic bodies

Minor points

  • Introduction: While the introduction is concise, a few more details could be added to help improve its readability and to rationalise the context of this study within the current literature. It is also important to note that not all readers have background knowledge of pSS, EVs or proteomics, so sufficient (and concise) background should be provided.
    1. Very little description has been provided for primary sjogren’s syndrome (pSS) to rationalise why a biomarker study is important.
    2. Could the authors provide a very brief overview of why SWATH-MS was selected to analyse the EV proteomes for this study? SWATH-MS is a data independent method that provides an advantage over data-dependent mass spectrometry methods as it is more reliable and can improve protein identifications.
    3. Maintain consistency of pSS abbreviation. In some passages, SS is used.
  • Results/Figures/Tables:
    1. Provide a description of the spectral library, in particular the number of proteins that comprised the library. This will help put into perspective the proteomic coverage that was achieved when the SWATH-MS data for EV and WS was aligned to the spectral library
    2. Figure 2F: provide the p-values on the figure next to each bar or in the figure legend
  • Methods:
    1. if the study had 15 patients and 15 healthy controls, why were only a portion of them used for the analysis? Based on section 4.2 described results, I have assumed that the cohorts are pSS (n=7) and healthy controls (n=5). Please rectify and clarify the samples that were used for each comparison.
    2. Were the healthy controls age/gender matched?
    3. Can more information be provided on the saliva sample collection (food/drink restrictions, time of collection, volume of saliva, etc)?
    4. It is unclear if both EVs and WS proteomes were assessed from each individual patient and healthy control. Were the EVs and WS that were compared derived from the same saliva samples? Please clarify

Author Response

Dear Referee’s,

Please see below for my responses to your comments and the revised manuscript with the changes, highlighted in yellow.

Comments to the Author:

The overarching goal of this study is to characterise the proteome of salivary extracellular vesicles (EVs) and whole saliva derived primary sjogren’s syndrome (pSS) patients. The proteomes were assessed by SWATH-mass spectrometry, a data-independent mass spectrometry approach that allows for reliable and targeted proteomic analyses. SWATH-MS is growing as a useful proteomic method biomarker discovery as SWATH-MS theoretically sequences all analyses within a sample, and the data can be mined retrospectively as the spectral library grows. The study identified a number of differentially expressed proteins between pSS and healthy control EVs that were related to the immune response. This is an interesting study and holds immense value for developing EV-based biomarker panels that can be used to diagnose and monitor autoimmune disorders such as pSS.

The methodological approach for the manuscript is logical, however there are a few key points that should be addressed to improve the content of the manuscript. Furthermore, several passages would benefit from further refinement and clarification to improve the readability of the manuscript. Please refer to the major and minor points below.

Response: We appreciate the reviewer’s point that the manuscript is logical but that it needs some refinement and clarification that will improve the readability. We have clarified the issues in the manuscript and answered to all requests.

Major points:

As this is a study centred around extracellular vesicles (EV), it is imperative that the authors follow the MISEV2018 guidelines for the characterisation of their salivary EV populations. Is there a reason that the authors have omitted from displaying an EV characterisation figure in this study? EV studies typically show the quality, size and content of their EV populations in a figure. Typically, such a figure would comprise of a size distribution profile determined by nanoparticle tracking, EV quality/integrity/size determined by transmission electron microscopy and the identification of EV-marker proteins determined by western blotting or mass spectrometric methods. I would recommend the authors to revise the MISEV2018 guidelines and provide a figure containing the appropriate characterisations for their isolated salivary EV populations.

Response: Thank you for pointing this out, it is an important issue and, as stated by the reviewer it is imperative to follow the MISEV2018 guidelines. Nevertheless, MISEV guidelines can be well applied when the scope is to isolate a specific population of EVs. In this manuscript, UC is used to enrich EVs from soluble proteins in saliva samples. A Dynamic Light scattering (DLS) analysis on Saliva enriched EVs was performed and the result is reported in figure S1. The description of the technique is in MM in the revised manuscript (new paragraph 4.4). Moreover, the obtained library was compared with Vesiclepedia (TOP100 most common human EV proteins) database and 35 proteins resulted comprised in this database (Supplemental table 1).

DLS results demonstrate the broad size of our EVs and the definition in the paper was changed in “EV-enriched saliva”

Based on these characterisations, it is recommended that the broad term of EVs is used and that EVs are categorised by their size ranges (eg., small-EVs, large EVs, etc) rather than exosomes/microparticles/apoptotic bodies

Response: DLS results demonstrate the broad size of our EVs and the definition in the paper was changed in “EV-enriched saliva”

Minor points:

Introduction: 

While the introduction is concise, a few more details could be added to help improve its readability and to rationalise the context of this study within the current literature. It is also important to note that not all readers have background knowledge of pSS, EVs or proteomics, so sufficient (and concise) background should be provided.

Very little description has been provided for primary Sjogren’s syndrome (pSS) to rationalise why a biomarker study is important.

Response: According to referee’s suggestions we have added some more information in the introduction. See the modified version …. of the revised manuscript (page 1 and 2) New references have been added.

Could the authors provide a very brief overview of why SWATH-MS was selected to analyse the EV proteomes for this study? SWATH-MS is a data independent method that provides an advantage over data-dependent mass spectrometry methods as it is more reliable and can improve protein identifications.

Response: We clarified this point in the introduction (page 2): The advantages in using this data-independent acquisition method rely on its unique feature to combine the deep proteome coverage capabilities of typical shotgun proteomics with accurate quantification of targeted proteomics without suffering of their own limitations (a random and irreproducible precursor ion selection that lead to under-sampling and the lack in quantifying large fractions of a proteome due to the limited number of transitions to measure, respectively).

Maintain consistency of pSS abbreviation. In some passages, SS is used.

Response: Thank you for this observation, we checked abbreviations throughout the manuscript.

Results/Figures/Tables:

Provide a description of the spectral library, in particular the number of proteins that comprised the library. This will help put into perspective the proteomic coverage that was achieved when the SWATH-MS data for EV and WS was aligned to the spectral library

Response: A list of identified and quantified proteins in EVs and WS of healthy controls, a list of protein abundances, means, fold changes and p-values of EVs from healthy controls and pSS patients and the list of differential expression proteins in pSS (|FC|>1.5) were reported in Tables S1, S2 and S3A.

Figure 2F: provide the p-values on the figure next to each bar or in the figure legend

Response: we have appreciated the comment of the reviewer, however we prefer to not include the p-values besides each bar of figure 2F because those values are already inherent in the figure (X-axis). Moreover, we highlighted the values within the text (line 196-199).

Methods:

if the study had 15 patients and 15 healthy controls, why were only a portion of them used for the analysis? Based on section 4.2 described results, I have assumed that the cohorts are pSS (n=7) and healthy controls (n=5). Please rectify and clarify the samples that were used for each comparison.

Response: Thank you for your comment, indeed we collected and quantified by BCA the salivary WS proteomes of 15 patients and 15 healthy controls whereas we were able to analyze the EV proteomic profiles of 7 and 5 controls. We clarified this in the manuscript (page 10)

Were the healthy controls age/gender matched?

Response: We confirm that they were matched.

Can more information be provided on the saliva sample collection (food/drink restrictions, time of collection, volume of saliva, etc)?

Response: All the information requested was added to Material and methods in the revised manuscript  (new 4.2 paragraph)

It is unclear if both EVs and WS proteomes were assessed from each individual patient and healthy control. Were the EVs and WS that were compared derived from the same saliva samples? Please clarify

Response: Thank you for this comment and we agree that this point needs to be clarified. EV and WS proteomes were assessed from individual healthy control and comparison derived from the same saliva samples split immediately after collection. We added a clarifying sentence on pag 10 in the revised manuscript.

Reviewer 2 Report

The study explore the content of proteins in whole saliva and isolated EVs form saliva from pSS patients and with whole saliva and EVs from healthy persons. It is an interesting and well written manuscript where the proteomic part is well covered. Characterization of the EV preps should be  been better described by identifying EV markers and TEM image to assure for presence of adequat EVs

  • The introduction state in line 40…”that EV have been extensively analyzed in pSS patients to characterize the expression of small RNA molecules. However, little is known about the protein composition” … lack of references to both of these statements
  • The procedure of saliva sampling and volumes obtained from patients and healthy persons are not described.
  • The purification of EVs from saliva is performed by differential ultracentrifugation. Volume used is not mentioned.
  • The EV preparations are not characterized due to Minimal for studies of extracellular vesicles (MISEV) 2018

Author Response

Dear Referee’s,

Please see below for my responses to your comments and the revised manuscript with the changes, highlighted in yellow.

Comments to the Author:

The study explore the content of proteins in whole saliva and isolated EVs form saliva from pSS patients and with whole saliva and EVs from healthy persons. It is an interesting and well written manuscript where the proteomic part is well covered. Characterization of the EV preps should be  been better described by identifying EV markers and TEM image to assure for presence of adequat EVs

Response: Thank you for this nice comment. We also appreciate the emphasis on the necessity of a better characterization of EV preps. We faced this point in particular below.

The introduction state in line 40…”that EV have been extensively analyzed in pSS patients to characterize the expression of small RNA molecules. However, little is known about the protein composition” … lack of references to both of these statements

Response: thank you for underlining the lack of references. We modified the introduction according to your suggestions also quoting additional references (citations  in the introduction page 1)

The procedure of saliva sampling and volumes obtained from patients and healthy persons are not described.

Response: Thank you for pointing this out, lacking information was added in Materials and Methods in the revised manuscript. on pages 9-10

The purification of EVs from saliva is performed by differential ultracentrifugation. Volume used is not mentioned.

Response: We added requested data in Materials and Methods in the revised manuscript version. on page 10

The EV preparations are not characterized due to Minimal for studies of extracellular vesicles (MISEV) 2018

Response: Thank you for pointing this out, it is an important issue and, as stated by the reviewer it is imperative to follow the MISEV2018 guidelines. Nevertheless, MISEV guidelines can be well applied when the scope is to isolate a specific population of EVs. In this manuscript, UC is used to enrich EVs from soluble proteins in saliva samples. A Dynamic Light scattering (DLS) analysis on Saliva enriched EVs was performed and the result is reported in figure S1. The description of the technique is in MM in the revised manuscript (new paragraph 4.4). Moreover, the obtained library was compared with Vesiclepedia (TOP100 most common human EV proteins) database and 35 proteins resulted comprised in this database (Supplemental table 1).

Round 2

Reviewer 1 Report

The overarching goal of this study is to characterise the proteome of salivary extracellular vesicles (EVs) and whole saliva derived primary sjogren’s syndrome (pSS) patients. The proteomes were assessed by SWATH-mass spectrometry, a data-independent mass spectrometry approach that allows for reliable and targeted proteomic analyses. SWATH-MS is growing as a useful proteomic method biomarker discovery as SWATH-MS theoretically sequences all analyses within a sample, and the data can be mined retrospectively as the spectral library grows. The study identified a number of differentially expressed proteins between pSS and healthy control EVs that were related to the immune response. This is an interesting study and holds immense value for developing EV-based biomarker panels that can be used to diagnose and monitor autoimmune disorders such as pSS.

The methodological approach for the manuscript is logical, however there are a few key points that should be addressed to improve the content of the manuscript. Furthermore, several passages would benefit from further refinement and clarification to improve the readability of the manuscript. Please refer to the major and minor points below.

Response: We appreciate the reviewer’s point that the manuscript is logical but that it needs some refinement and clarification that will improve the readability. We have clarified the issues in the manuscript and answered to all requests.

Major points:

As this is a study centred around extracellular vesicles (EV), it is imperative that the authors follow the MISEV2018 guidelines for the characterisation of their salivary EV populations. Is there a reason that the authors have omitted from displaying an EV characterisation figure in this study? EV studies typically show the quality, size and content of their EV populations in a figure. Typically, such a figure would comprise of a size distribution profile determined by nanoparticle tracking, EV quality/integrity/size determined by transmission electron microscopy and the identification of EV-marker proteins determined by western blotting or mass spectrometric methods. I would recommend the authors to revise the MISEV2018 guidelines and provide a figure containing the appropriate characterisations for their isolated salivary EV populations.

Response: Thank you for pointing this out, it is an important issue and, as stated by the reviewer it is imperative to follow the MISEV2018 guidelines. Nevertheless, MISEV guidelines can be well applied when the scope is to isolate a specific population of EVs. In this manuscript, UC is used to enrich EVs from soluble proteins in saliva samples. A Dynamic Light scattering (DLS) analysis on Saliva enriched EVs was performed and the result is reported in figure S1. The description of the technique is in MM in the revised manuscript (new paragraph 4.4). Moreover, the obtained library was compared with Vesiclepedia (TOP100 most common human EV proteins) database and 35 proteins resulted comprised in this database (Supplemental table 1).

DLS results demonstrate the broad size of our EVs and the definition in the paper was changed in “EV-enriched saliva”

 Reviewer response: Could the authors also add a brief sentence to describe the EV size distribution, in particular the modal and/or mean sizes, as determined by dynamic light scattering in Results section 2.1 (lines 83-93).

Line 89: “The obtained library was compared… in this database” please clarify this sentence further. By ‘library” are the authors referring to the spectral library that was used to align the SWATH data? Was the spectral library compared to the Top 100 EV proteins as reported by Vesiclepedia?  Or are the proteins listed in supplemental table 1 the extracted SWATH data for the healthy EVs (n=5)? If that is the case, then it would be more appropriate to start the sentence with something along the lines of “The proteins sequenced by SWATH-MS in the salivary EVs were compared with Vesiclepedia top 100 EV proteins…”

On further inspection of supplemental table 2, it appears that EV-marker proteins such as CD9 and PDCD6I have been identified and quantified in the salivary EVs. However, these proteins are not listed in supplementary table 1 library proteins. Were these proteins in the spectral library, so that they were identified in the SWATH-MS analyses of the EV samples? If so, why are they not listed in supplementary table 1? It would be good to make note in the manuscript text that typical EV-marker proteins (list 3, including CD9 and PDCD6I) have been identified in the salivary EVs. Please rectify

TEM images of the salivary EVs/ enriched EVs is recommended.  Adequate characterisation is foundational to an EV study to make any EV-associated conclusions.

Based on these characterisations, it is recommended that the broad term of EVs is used and that EVs are categorised by their size ranges (eg., small-EVs, large EVs, etc) rather than exosomes/microparticles/apoptotic bodies

Response: DLS results demonstrate the broad size of our EVs and the definition in the paper was changed in “EV-enriched saliva”

Reviewer response: As the size distribution profile of the salivary EVs is broad, could the authors also add a brief description in Results section 2.1 (lines 83-93) of the EV size distribution, in particular the modal and/or mean sizes, as determined by dynamic light scattering.

Minor points:

Introduction: 

While the introduction is concise, a few more details could be added to help improve its readability and to rationalise the context of this study within the current literature. It is also important to note that not all readers have background knowledge of pSS, EVs or proteomics, so sufficient (and concise) background should be provided.

Very little description has been provided for primary Sjogren’s syndrome (pSS) to rationalise why a biomarker study is important.

Response: According to referee’s suggestions we have added some more information in the introduction. See the modified version …. of the revised manuscript (page 1 and 2) New references have been added.

Could the authors provide a very brief overview of why SWATH-MS was selected to analyse the EV proteomes for this study? SWATH-MS is a data independent method that provides an advantage over data-dependent mass spectrometry methods as it is more reliable and can improve protein identifications.

Response: We clarified this point in the introduction (page 2): The advantages in using this data-independent acquisition method rely on its unique feature to combine the deep proteome coverage capabilities of typical shotgun proteomics with accurate quantification of targeted proteomics without suffering of their own limitations (a random and irreproducible precursor ion selection that lead to under-sampling and the lack in quantifying large fractions of a proteome due to the limited number of transitions to measure, respectively).

Maintain consistency of pSS abbreviation. In some passages, SS is used.

Response: Thank you for this observation, we checked abbreviations throughout the manuscript.

Results/Figures/Tables:

Provide a description of the spectral library, in particular the number of proteins that comprised the library. This will help put into perspective the proteomic coverage that was achieved when the SWATH-MS data for EV and WS was aligned to the spectral library

Response: A list of identified and quantified proteins in EVs and WS of healthy controls, a list of protein abundances, means, fold changes and p-values of EVs from healthy controls and pSS patients and the list of differential expression proteins in pSS (|FC|>1.5) were reported in Tables S1, S2 and S3A.

Reviewer response: In Supplemental table 1, are the listed proteins solely spectral library proteins as identified by IDA LC-MS/MS, or are these the proteins that were extracted by aligning the swath data to the spectral library? Please clarify

Figure 2F: provide the p-values on the figure next to each bar or in the figure legend

Response: we have appreciated the comment of the reviewer, however we prefer to not include the p-values besides each bar of figure 2F because those values are already inherent in the figure (X-axis). Moreover, we highlighted the values within the text (line 196-199).

Methods:

if the study had 15 patients and 15 healthy controls, why were only a portion of them used for the analysis? Based on section 4.2 described results, I have assumed that the cohorts are pSS (n=7) and healthy controls (n=5). Please rectify and clarify the samples that were used for each comparison.

Response: Thank you for your comment, indeed we collected and quantified by BCA the salivary WS proteomes of 15 patients and 15 healthy controls whereas we were able to analyze the EV proteomic profiles of 7 and 5 controls. We clarified this in the manuscript (page 10)

Reviewer response: what is meant by 7/15 pSS patients and 5/15 healthy controls? This is quite confusing. No BCA method/results have been provided in the manuscript to list a cohort comprised of 15 pSS patients and 15 healthy controls. Also, the proteomic data that comprises this whole manuscript, has only been captured for 7 pSS and 5 healthy controls. If that is correct, I would recommend that the authors only mention pSS (n=7) and healthy control (n=5) as their cohort, even though it is small.

Were the healthy controls age/gender matched?

Response: We confirm that they were matched.

 Reviewer response: I would recommend that the authors mention that the controls are age and gender matched in Section 4.1 (344-345). Line 345 says that the control saliva samples were derived from women. Were all the pSS patients also women?

Can more information be provided on the saliva sample collection (food/drink restrictions, time of collection, volume of saliva, etc)?

Response: All the information requested was added to Material and methods in the revised manuscript  (new 4.2 paragraph)

Reviewer response: Line 345  - indicate that the parameter for the healthy controls being described is an unstimulated salivary flow rate (mean 0.467 ± 0.299 ml/minute).

It is unclear if both EVs and WS proteomes were assessed from each individual patient and healthy control. Were the EVs and WS that were compared derived from the same saliva samples? Please clarify

Response: Thank you for this comment and we agree that this point needs to be clarified. EV and WS proteomes were assessed from individual healthy control and comparison derived from the same saliva samples split immediately after collection. We added a clarifying sentence on pag 10 in the revised manuscript.

 Reviewer response: Please revise this newly added text and fix any errors. i.e. ‘WA’ to ‘WS’, and ‘comparison’ to ‘pSS patients’.

Author Response

DLS results demonstrate the broad size of our EVs and the definition in the paper was changed in “EV-enriched saliva”

 Reviewer response: Could the authors also add a brief sentence to describe the EV size distribution, in particular the modal and/or mean sizes, as determined by dynamic light scattering in Results section 2.1 (lines 83-93).

As suggested, we inserted a brief description at the beginning of section 2.1 describing size and distribution of the three main components of our EV-enriched saliva. The same data are n reported along with figure S1 in supplementary material.

Line 89: “The obtained library was compared… in this database” please clarify this sentence further. By ‘library” are the authors referring to the spectral library that was used to align the SWATH data? Was the spectral library compared to the Top 100 EV proteins as reported by Vesiclepedia?  Or are the proteins listed in supplemental table 1 the extracted SWATH data for the healthy EVs (n=5)? If that is the case, then it would be more appropriate to start the sentence with something along the lines of “The proteins sequenced by SWATH-MS in the salivary EVs were compared with Vesiclepedia top 100 EV proteins…”

Response to the reviewer: The reviewer is totally right, there is a misleading in this sentence. In fact, in this section we won’t refer to the spectral library that has been used for SWATH MS data matching, but we want to describe the dataset of EVs proteins obtained by IDA MS (DDA) to compare with WS dataset. We apologize for the mistake and we have replaced the entire sentence with “The EVs dataset...”(referring to the 278 identified and quantified proteins in EVs fraction).

On further inspection of supplemental table 2, it appears that EV-marker proteins such as CD9 and PDCD6I have been identified and quantified in the salivary EVs. However, these proteins are not listed in supplementary table 1 library proteins. Were these proteins in the spectral library, so that they were identified in the SWATH-MS analyses of the EV samples? If so, why are they not listed in supplementary table 1? It would be good to make note in the manuscript text that typical EV-marker proteins (list 3, including CD9 and PDCD6I) have been identified in the salivary EVs. Please rectify

Response to the reviewer: Again, this comment is a consequence of our mistake in writing the word “library” at line 89, as the reviewer has reported above. Supplementary table 1 is not the spectral library used to match with the SWATH MS data, but the list of the identified/quantified proteins in EVs through the IDA (DDA) MS approach to be compared with the WS dataset. The spectral library used to analyze SWATH MS data will be provided in sptxt and splib format and uploaded in ProteomeXchange repository as soon as the manuscript will be accepted. Or, I can provide to the reviewer these files if the reviewer need further inspections.

TEM images of the salivary EVs/ enriched EVs is recommended.  Adequate characterisation is foundational to an EV study to make any EV-associated conclusions.

We apologize for not being able to accomplish to this request. Current lockdown and work restrictions in Italy due to COVID pandemic disease does not allow to provide these data within a reasonable timeframe (months). However, we wish to stress that our samples are quite complex, as shown by DLS analysis. This wide distribution would make TEM analysis not truly representative of the real complexity of the sample, since by its nature this technique can provide robust results on quite homogeneous samples (i.e. in our previous article Comelli L, Rocchiccioli S, Smirni S, Salvetti A, Signore G, Citti L, Trivella MG, Cecchettini A. Characterization of secreted vesicles from vascular smooth muscle cells. Mol Biosyst. 2014 May;10(5):1146-52. doi: 10.1039/c3mb70544g. PMID: 24626815.) with narrow size distribution.  Thus, a TEM analysis on a sample with vesicles spanning almost two orders of magnitude in size would likely be of little information for the reader, if not misleading. Additionally, while we adhere and generally endorse recommendations by MISEV2018, we think that the multimodal distribution of vesicles in our samples would provide only unreliable results if treated with the (otherwise fully reasonable) standardization approach. We added a sentence in the manuscript to evidence this fact. We thank the reviewer for his/her understanding.

Based on these characterisations, it is recommended that the broad term of EVs is used and that EVs are categorised by their size ranges (eg., small-EVs, large EVs, etc) rather than exosomes/microparticles/apoptotic bodies

Response: DLS results demonstrate the broad size of our EVs and the definition in the paper was changed in “EV-enriched saliva”

Reviewer response: As the size distribution profile of the salivary EVs is broad, could the authors also add a brief description in Results section 2.1 (lines 83-93) of the EV size distribution, in particular the modal and/or mean sizes, as determined by dynamic light scattering.

As suggested, we inserted a brief description at the beginning of section 2.1 describing size and distribution of the three main components of our EV-enriched saliva. The same data are now reported along with figure s1 in supplementary material.

Reviewer response: In Supplemental table 1, are the listed proteins solely spectral library proteins as identified by IDA LC-MS/MS, or are these the proteins that were extracted by aligning the swath data to the spectral library? Please clarify

Response to the reviewer: the identified/quantified proteins listed in Supplemental table 1 derive from the shotgun proteomics analysis IDA (DDA) MS approach used to compare EVs and WS from healthy controls. These datasets are not those ones used to analyze SWATH MS data. Indeed, spectral library has been created by first pooling together EVs samples from healthy controls and pSS patients and WS samples from healthy controls (Material and methods, paragraph 4.6). Pooled samples were then injected in 4 technical replicates and analyzed by IDA MS method in order to increase the identification rate.

Reviewer response: what is meant by 7/15 pSS patients and 5/15 healthy controls? This is quite confusing. No BCA method/results have been provided in the manuscript to list a cohort comprised of 15 pSS patients and 15 healthy controls. Also, the proteomic data that comprises this whole manuscript, has only been captured for 7 pSS and 5 healthy controls. If that is correct, I would recommend that the authors only mention pSS (n=7) and healthy control (n=5) as their cohort, even though it is small.

The reviewer is right. We have replaced the sentence “7/15 pSS patients and 5/15 healthy controls” with the exact number of biological replicates used in this study and thus 7 pSS patients and 5 healthy controls (4.1 and 4.3 paragraphs).

 Reviewer response: I would recommend that the authors mention that the controls are age and gender matched in Section 4.1 (344-345). Line 345 says that the control saliva samples were derived from women. Were all the pSS patients also women?

Yes all pSS patients are women, Sjogren Syndrome is occurs predominately in women over men (16:1).

Can more information be provided on the saliva sample collection (food/drink restrictions, time of collection, volume of saliva, etc)?

Response: All the information requested was added to Material and methods in the revised manuscript  (new 4.2 paragraph)

Reviewer response: Line 345  - indicate that the parameter for the healthy controls being described is an unstimulated salivary flow rate (mean 0.467 ± 0.299 ml/minute).

Thank you for the observation, we added the information in the text.

Reviewer 2 Report

Accept in present form

Author Response

Thank you for the comments

Round 3

Reviewer 1 Report

Comments have been added to the attached word doc.(Review 3)- reviewer responses are in italicised and underlined writing with grey background

Reviewer response: thank you for clarifying. Yet, this section in the manuscript is still unclear as it currently states on line 188 that the proteomic analysis is done by SWATH-MS. If the proteins for WS and EVs in section 2.1 were obtained by IDA (as stated here and below) then please change this in the text. It currently states that the presented data in section 2.1 is SWATH-MS proteomic data and that is quite confusing. Is supplemental table 1 the IDA data, and supplemental table 2, the SWATH data? It would be good to clarify this information in the supplementary materials legends on page 13(lines 1065-1075) and in the respective texts in sections 2.1 and 2.2

Response to the reviewer: we would like to thank the reviewer for highlighting this writing mistake. The sentence has been modified in the text (section 2.1) and a reference that data derived from SWATH-MS has been added in section 2.2. Moreover, the data derivation has been included in the supplemental table 1 and 2 legends on page 13 and on the supplemental tables 1 and 2 excel files, as well.

On further inspection of supplemental table 2, it appears that EV-marker proteins such as CD9 and PDCD6I have been identified and quantified in the salivary EVs. However, these proteins are not listed in supplementary table 1 library proteins. Were these proteins in the spectral library, so that they were identified in the SWATH-MS analyses of the EV samples? If so, why are they not listed in supplementary table 1? It would be good to make note in the manuscript text that typical EV-marker proteins (list 3, including CD9 and PDCD6I) have been identified in the salivary EVs. Please rectify

Response to the reviewer: Again, this comment is a consequence of our mistake in writing the word “library” at line 89, as the reviewer has reported above. Supplementary table 1 is not the spectral library used to match with the SWATH MS data, but the list of the identified/quantified proteins in EVs through the IDA (DDA) MS approach to be compared with the WS dataset. The spectral library used to analyze SWATH MS data will be provided in sptxt and splib format and uploaded in ProteomeXchange repository as soon as the manuscript will be accepted. Or, I can provide to the reviewer these files if the reviewer need further inspections.

Reviewer response: depositing the spectral library to ProteomeXchange is sufficient

Response to the reviewer: we agree with the reviewer.

TEM images of the salivary EVs/ enriched EVs is recommended.  Adequate characterisation is foundational to an EV study to make any EV-associated conclusions.

We apologize for not being able to accomplish to this request. Current lockdown and work restrictions in Italy due to COVID pandemic disease does not allow to provide these data within a reasonable timeframe (months). However, we wish to stress that our samples are quite complex, as shown by DLS analysis. This wide distribution would make TEM analysis not truly representative of the real complexity of the sample, since by its nature this technique can provide robust results on quite homogeneous samples (i.e. in our previous article Comelli L, Rocchiccioli S, Smirni S, Salvetti A, Signore G, Citti L, Trivella MG, Cecchettini A. Characterization of secreted vesicles from vascular smooth muscle cells. Mol Biosyst. 2014 May;10(5):1146-52. doi: 10.1039/c3mb70544g. PMID: 24626815.) with narrow size distribution.  Thus, a TEM analysis on a sample with vesicles spanning almost two orders of magnitude in size would likely be of little information for the reader, if not misleading. Additionally, while we adhere and generally endorse recommendations by MISEV2018, we think that the multimodal distribution of vesicles in our samples would provide only unreliable results if treated with the (otherwise fully reasonable) standardization approach. We added a sentence in the manuscript to evidence this fact. We thank the reviewer for his/her understanding.

 Reviewer response: Please revise this newly added text and fix any errors. i.e. ‘WA’ to ‘WS’, and ‘comparison’ to ‘pSS patients’.

Reviewer response: Page 10, Line 765 (Methods 4.3) “EVs and WS proteomes were assessed from each individual healthy control and proteomics data…). EVs and WS were assessed from both pSS and healthy controls? Please edit this sentence to make it clear, otherwise it currently appears that only healthy controls were processed and analyzed.

Response to the reviewer: we would like to thank the reviewer to have pinpointed this unclear sentence. The sentence has been completely modified accordingly.

Thank you. We fixed the errors

Additional point:

Can the authors indicate which samples are pSS and controls (and their replicates) in Supplemental Tables 1 and 2

Response to the reviewer: we have now included in both supplemental tables 1 and 2 the heads that specify the name of each group and replicate